# Universally Quantized Neural Compression

**Eirikur Agustsson**[*]
Google Research
eirikur@google.com

**Lucas Theis**[*]
Google Research
theis@google.com

## Abstract

A popular approach to learning encoders for lossy compression is to use additive uniform noise during training as a differentiable approximation to test-time quantization. We demonstrate that a uniform noise channel can also be implemented at test time using *universal quantization* (Ziv, 1985). This allows us to eliminate the mismatch between training and test phases while maintaining a completely differentiable loss function. Implementing the uniform noise channel is a special case of the more general problem of communicating a sample, which we prove is computationally hard if we do not make assumptions about its distribution. However, the uniform special case is efficient as well as easy to implement and thus of great interest from a practical point of view. Finally, we show that quantization can be obtained as a limiting case of a soft quantizer applied to the uniform noise channel, bridging compression with and without quantization.

## 1 Introduction

Over the last four years, deep learning research into lossy image compression has seen tremendous progress. End-to-end trained neural networks have gone from barely beating JPEG2000 [4] to outperforming the best manually designed compression schemes for images [36, 2]. Despite this success, many challenges remain before end-to-end trained compression becomes a viable alternative to more traditional codecs. Computational complexity, temporal inconsistencies, and perceptual metrics which are effective yet easy to optimize are some of the challenges facing neural networks.

In this paper we focus on the issue of quantization. Practical lossy compression schemes rely on quantization to compute a discrete representation which can be transmitted digitally. But quantization is a non-differentiable operation and as such prevents us from optimizing encoders directly via backpropagation [33]. A common workaround is to replace quantization with a differentiable approximation during training but to use quantization at test time [e.g., 32, 4, 1]. However, it is unclear how much this mismatch between training and test phases is hurting performance.

A promising alternative is to get rid of quantization altogether [15]. That is, to communicate information in a differentiable manner both at training and at test time. At the heart of this approach is the insight that we can communicate a sample from a possibly continuous distribution using a finite number of bits, also known as the *reverse Shannon theorem* [8]. However, existing realizations of this approach tend to be either computationally costly or statistically inefficient, that is, they require more bits than they transmit information.

Here, we bridge the gap between the two approaches of dealing with quantization. A popular approximation for quantization is additive uniform noise [4, 5]. In Section 3.2, we show that additive uniform noise can be viewed as an instance of compression without quantization and describe a technique for implementing it at test time. Unlike other approaches to quantizationless compression, this technique is both statistically and computationally efficient. In Section 4.1, we show how to smoothly interpolate between uniform noise and hard quantization while maintaining differentiability.

---

[*]Equal contribution

We further show that it is possible to analytically integrate out noise when calculating gradients and in some cases drastically reduce their variance (Section 4.2). Finally, we evaluate our approach empirically in Section 5 and find that a better match between training and test phases leads to improved performance especially in models of lower complexity.

## 2 Related work

Most prior work on end-to-end trained lossy compression optimizes a rate-distortion loss of the form

$$-\log_2 P(\lfloor f(\boldsymbol{x}) \rceil) + \lambda d(\boldsymbol{x}, g(\lfloor f(\boldsymbol{x}) \rceil)). \tag{1}$$

Here, $f$ is an encoder, $g$ is a decoder, $P$ is a probability mass function and they may all depend on parameters we want to optimize. The distortion $d$ measures the discrepancy between inputs and reconstructions and the parameter $\lambda > 0$ controls the trade-off between it and the number of bits. The rounding function $\lfloor \cdot \rceil$ used for quantization and the discreteness of $P$ pose challenges for optimizing the encoder.

Several papers have proposed methods to deal with quantization for end-to-end trained lossy compression. Toderici et al. [32] replaced rounding with stochastic rounding to the nearest integer. Theis et al. [31] applied hard quantization during both training and inference but used straight-through gradient estimates to obtain a training signal for the encoder. Agustsson et al. [1] used a smooth approximation of vector quantization that was annealed towards hard quantization during training.

Most relevant for our work is the approach taken by Ballé et al. [4], who proposed to add uniform noise during training,

$$-\log_2 p(f(\boldsymbol{x}) + \boldsymbol{u}) + \lambda d(\boldsymbol{x}, g(f(\boldsymbol{x}) + \boldsymbol{u})), \tag{2}$$

as an approximation to rounding at test time. Here, $p$ is a density and $\boldsymbol{u}$ is a sample of uniform noise drawn from $U([-0.5, 0.5)^D)$. If the distortion is a mean-squared error, then this approach is equivalent to a variational autoencoder [25, 18] with a uniform encoder [5, 31].

Another line of research studies the simulation of noisy channels using a noiseless channel, that is, the reverse of *channel coding*. In particular, how can we communicate a sample $\boldsymbol{z}$ from a conditional distribution (the noisy channel), $q(\boldsymbol{z} \mid \boldsymbol{x})$, using as few bits as possible (the noiseless channel)? The reverse Shannon theorem of Bennett and Shor [8] shows that it is possible to communicate a sample using a number of bits not much larger than the mutual information between $\boldsymbol{X}$ and $\boldsymbol{Z}$, $I[\boldsymbol{X}, \boldsymbol{Z}]$.

Existing implementations of reverse channel coding operate on the same principle. First, a large number of samples $\boldsymbol{z}_n$ is generated from a fixed distribution $p$. Importantly, this distribution does not depend on $\boldsymbol{x}$ and the same samples can be generated on both the sender's and the receiver's side using a shared source of randomness (for our purposes this would be a pseudorandom number generator with a fixed seed). One of these samples is then selected and its index $n$ communicated digitally. The various methods differ in how this index is selected.

Cuff [11] provided a constructive achievability proof for the mutual information bound using an approach which was later dubbed the *likelihood encoder* [12]. In this approach the index $n$ is picked stochastically with a probability proportional to $p(\boldsymbol{x} \mid \boldsymbol{z}_n)$. An equivalent approach dubbed MIRACLE was later derived by Havasi et al. [15] using importance sampling. In contrast to Cuff and Song [12], Havasi et al. [15] considered communication of a single sample from $q$ instead of a sequence of samples. MIRACLE also represents the first application of quantizationless compression in the context of neural networks. Originally designed for model compression, it was recently adapted to the task of lossy image compression [13]. An earlier but computationally more expensive method based on rejection sampling was described by Harsha et al. [14].

Li and Gamal [21] described a simple yet efficient approach. The authors proved that it uses at most

$$I[\boldsymbol{X}, \boldsymbol{Z}] + \log_2(I[\boldsymbol{X}, \boldsymbol{Z}] + 1) + 4 \tag{3}$$

bits on average. To our knowledge, this is the lowest known upper bound on the bits required to communicate a single sample. The overhead is still significant if we want to communicate a small amount of information but becomes negligible as the mutual information increases.

Finally, we will rely heavily on results on uniform dither and *universal quantization* [29, 37, 35] to communicate a sample from a uniform distribution (Section 3.2). Choi et al. [9] used universal quantization as a relaxation of hard quantization. However, universal quantization was used in a

manner that still produced a non-differentiable loss, which the authors dealt with by using straight-through gradient estimates [7]. In contrast, here we will use fully differentiable losses during training and use the same method of encoding at training and at test time. Roberts [27] applied universal quantization directly to grayscale pixels and found it lead to superior picture quality compared to quantization.

## 3  Compression without quantization

Instead of approximating quantization or relying on straight-through gradient estimates, we would like to use a differentiable channel and thus eliminate any need for approximations during training. Existing methods to simulate a noisy channel $q_{\boldsymbol{Z}|\boldsymbol{x}}$ require simulating a number of random variables $\boldsymbol{Z}_n \sim p_{\boldsymbol{Z}}$ which is exponential in $D_{\mathrm{KL}}[q_{\boldsymbol{Z}|\boldsymbol{x}} \,||\, p_{\boldsymbol{Z}}]$ for every $\boldsymbol{x}$ we wish to communicate [e.g., 15].

Since the mutual information $I[\boldsymbol{X}, \boldsymbol{Z}]$ is a lower bound on the average Kullback-Leibler divergence, this creates a dilemma. On the one hand, we would like to keep the divergence small to limit the computational cost. For example, by encoding blocks of coefficients (sometimes also referred to as "latents") separately [15, 13]. On the other hand, the information transmitted should be large to keep the statistical overhead small (Equation 3).

One might hope that more efficient algorithms exist which can quickly identify an index $n$ without having to explicitly generate all samples. However, such an algorithm is not possible as it would allow us to efficiently sample distributions which are known to be hard to simulate even approximately (in terms of *total variation distance*, $D_{\mathrm{TV}}$) [22]. More precisely, we have the following lemma.

**Lemma 1.** *Consider an algorithm which receives a description of an arbitrary probability distribution $q$ as input and is also given access to an unlimited number of i.i.d. random variables $\boldsymbol{Z}_n \sim p$. It outputs $\boldsymbol{Z} \sim \tilde{q}$ such that its distribution is approximately $q$ in the sense that $D_{TV}[\tilde{q}, q] \leq 1/12$. If $RP \neq NP$, then there is no such algorithm whose time complexity is polynomial in $D_{KL}[q \,||\, p]$.*

A proof and details are provided in Appendix B. In order to design efficient algorithms for communicating samples, the lemma implies we need to make assumptions about the distributions involved.

### 3.1  Uniform noise channel

A particularly simple channel is the additive uniform noise channel,

$$\boldsymbol{Z} = f(\boldsymbol{x}) + \boldsymbol{U}, \quad \boldsymbol{U} \sim U([-0.5, 0.5)^D). \tag{4}$$

Replacing quantization with uniform noise during training is a popular strategy for end-to-end trained compression [e.g., 4, 5, 36]. In the following, however, we are no longer going to view this as an approximation to quantization but as a differentiable channel for communicating information. The uniform noise channel turns out to be easy to simulate computationally and statistically efficiently.

### 3.2  Universal quantization

For a fixed $y \in \mathbb{R}$, universal quantization is quantization with a random offset,

$$\lfloor y - U \rceil + U, \quad U \sim U([-0.5, 0.5)). \tag{5}$$

This form of quantization has the remarkable property of being equal in distribution to adding uniform noise directly [27, 29, 37]. That is,

$$\lfloor y - U \rceil + U \sim y + U', \tag{6}$$

where $U'$ is another source of identical uniform noise. This property has made universal quantization a useful tool for studying quantization, especially in settings where quantization noise $Y - \lfloor Y \rceil$ is roughly uniform. Here, we are interested in it not as an approximation but as a way to simulate a differentiable channel for communicating information. At training time, we will add uniform noise as in prior work [4, 5]. For deployment, we propose to use universal quantization instead of switching to hard quantization, thereby eliminating the mismatch between training and test phases.

If $Y$ is a random variable representing a coefficient produced by a transform, the encoder calculates discrete $K = \lfloor Y - U \rceil$ and transmits it to the decoder. The decoder has access to $U$ and computes $K + U$. How many bits are required to encode $K$? Zamir and Feder [35] showed that the conditional entropy of $K$ given $U$ is

$$H[K \mid U] = I[Y, Y + U] = h[Y + U]. \tag{7}$$

This bound on the coding cost has two important properties. First, being equivalent to the differential entropy of $Y + U$ means it is differentiable if the density of $Y$ is differentiable. Second, the cost of transmitting $K$ is equivalent to the amount of information gained by the decoder. In contrast to other methods for compression without quantization (Equation 3), the number of bits required is only bounded by the amount of information transmitted. In practice, we will use a model to approximate the distribution of $Y + U$ from which the distribution of $K$ can be derived, $P(K = k \mid U = u) = p_{Y+U}(k + u)$. Here, $p_{Y+U}$ is the same density that occurs in the loss in Equation 2.

Another advantage of universal quantization over more general reverse channel coding schemes is that it is much more computationally efficient. Its computational complexity grows only linearly with the number of coefficients to be transmitted instead of exponentially with the number of bits.

Universal quantization has previously been applied to neural networks using the same shift for all coefficients, $U_i = U_j$ [9]. We note that this form of universal quantization is not equivalent to adding either dependent or independent noise during training. Adding dependent noise would not create an information bottleneck, since a single coefficient which is always zero could be used by the decoder to recover the noise and therefore the exact values of the other coefficients. In the following, we will always assume independent noise as in Equation 4.

Generalizations to other forms of noise such as Gaussian noise are possible and are discussed in Appendix C. Here, we will focus on a simple uniform noise channel (Section 3.2) as frequently used in the neural compression literature [4, 5, 23, 36].

# 4 Compression with quantization

While the uniform noise channel has the advantage of being differentiable, there are still scenarios where we may want to use quantization. For instance, under some conditions universal quantization is known to be suboptimal with respect to mean squared error (MSE) [34, Theorem 5.5.1]. However, this assumes a fixed encoder and decoder. In the following, we show that quantization is a limiting case of universal quantization if we allow flexible encoders and decoders. Hence it is possible to recover any benefits quantization might have while maintaining a differentiable loss function.

## 4.1 Simulating quantization with uniform noise

We first observe that applying rounding as the last step of an encoder and again as the first step of a decoder would eliminate the effects of any offset $u \in [-0.5, 0.5)$,

$$\lfloor \lfloor y \rceil + u \rceil = \lfloor y \rceil. \tag{8}$$

This suggests that we may be able to recover some of the benefits of hard quantization without sacrificing differentiability by using a smooth approximation to rounding,

$$s(s(y) + u) \approx \lfloor y \rceil. \tag{9}$$

We are going to use the following function which is differentiable everywhere (Appendix C):

$$s_\alpha(y) = \lfloor y \rfloor + \frac{1}{2} \frac{\tanh(\alpha r)}{\tanh(\alpha/2)} + \frac{1}{2}, \quad \text{where} \quad r = y - \lfloor y \rfloor - \frac{1}{2}. \tag{10}$$

The function is visualized in Figure 1A. Its parameter $\alpha$ controls the fidelity of the approximation:

$$\lim_{\alpha \to 0} s_\alpha(y) = y, \quad \lim_{\alpha \to \infty} s_\alpha(y) = \lfloor y \rceil. \tag{11}$$

After observing a value $z$ for random variable $s_\alpha(Y) + U$, we can do slightly better if our goal is to minimize the MSE of $Y$. Instead of soft rounding twice, the optimal reconstruction is obtained with

$$r_\alpha(s_\alpha(y) + u), \quad \text{where} \quad r_\alpha(z) = \mathbb{E}[Y \mid s_\alpha(Y) + U = z]. \tag{12}$$

It is not difficult to see that

$$p(y \mid z) \propto \delta\left(y \in (s_\alpha^{-1}(z - 0.5), s_\alpha^{-1}(z + 0.5)]\right) p(y), \tag{13}$$

where $\delta$ evaluates to 1 if its argument is true and 0 otherwise. That is, the posterior over $y$ is a truncated version of the prior distribution. If we assume that the prior is smooth enough to be approximately uniform in each interval, we have

$$\mathbb{E}[Y \mid s_\alpha(Y) + U = z] \approx \frac{s_\alpha^{-1}(z - 0.5) + s_\alpha^{-1}(z + 0.5)}{2} = s_\alpha^{-1}(z - 0.5) + 0.5. \tag{14}$$

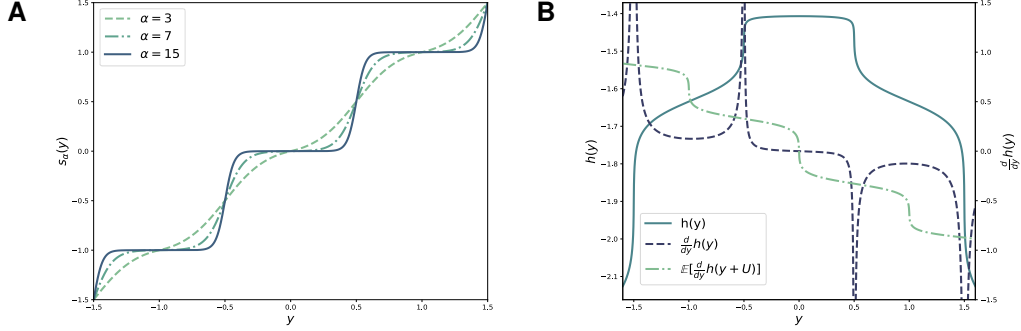

Figure 1: **A**: Visualization of the soft rounding function $s_\alpha$ for different choices of $\alpha$. **B**: Derivatives and expected derivatives for the log-density $h$ of a soft-rounded random variable $s_\alpha(X) + U$, where $X$ is assumed to have a logistic distribution.

where we have used that $s_\alpha(z+1) = s_\alpha(z) + 1$. We will assume this form for $r_\alpha$ going forward for which we still have that

$$\lim_{\alpha \to \infty} r_\alpha(s_\alpha(y) + u) = \lfloor y \rceil, \tag{15}$$

that is, we recover hard quantization as a limiting case. Thus in cases where quantization is desirable, we can anneal $\alpha$ towards hard quantization during training while still having a differentiable loss.

Smooth approximations to quantization have been used previously though without the addition of noise [1]. Note that soft rounding without noise does not create a bottleneck since the function is invertible and the input coefficients can be fully recovered by the decoder. Thus, Equation 15 offers a more principled approach to approximating quantization.

## 4.2 Reducing the variance of gradients

When $\alpha$ is large, the derivatives of $s_\alpha$ and $r_\alpha$ tend to be close to zero with high probability and very large with low probability. This leads to gradients for the encoder with potentially large variance. To compensate we propose to analytically integrate out the uniform noise as follows.

Let $h : \mathbb{R} \to \mathbb{R}$ be a differentiable function and, as before, let $U \sim U([-0.5, 0.5))$ be a uniform random variable. We are interested in computing the following derivative:

$$\frac{d}{dy} \mathbb{E}[h(y + U)] = \mathbb{E}\left[\frac{d}{dy} h(y + U)\right]. \tag{16}$$

To get a low-variance estimate of the expectation's derivative we could average over many samples of $U$. However, note that we also have

$$\frac{d}{dy} \mathbb{E}[h(y + U)] = \frac{d}{dy} \int_{y-0.5}^{y+0.5} h(y + u) du = h(y + 0.5) - h(y - 0.5). \tag{17}$$

That is, the gradient of the expectation can be computed analytically with finite differences. Furthermore, Equation 17 allows us to evaluate the derivative of the expectation even when $h$ is not differentiable.

Now consider the case where we apply $h$ pointwise to a vector $\boldsymbol{y} + \boldsymbol{U}$ with $\boldsymbol{U} \sim U([-0.5, 0.5)^D)$ followed by a multivariable function $\ell : \mathbb{R}^D \to \mathbb{R}$. Then

$$\frac{\partial}{\partial y_i} \mathbb{E}\left[\ell(h(\boldsymbol{y} + \boldsymbol{U}))\right] = \mathbb{E}\left[\left.\frac{\partial}{\partial z_i}\ell(\boldsymbol{Z})\right|_{\boldsymbol{Z}=h(\boldsymbol{y}+\boldsymbol{U})} \cdot \frac{\partial}{\partial y_i} h(y_i + U_i)\right] \tag{18}$$

$$\approx \mathbb{E}\left[\left.\frac{\partial}{\partial z_i}\ell(\boldsymbol{Z})\right|_{\boldsymbol{Z}=h(\boldsymbol{y}+\boldsymbol{U})}\right] \cdot \mathbb{E}\left[\frac{\partial}{\partial y_i} h(y_i + U_i)\right] \tag{19}$$

$$= \mathbb{E}\left[\left.\frac{\partial}{\partial z_i}\ell(\boldsymbol{Z})\right|_{\boldsymbol{Z}=h(\boldsymbol{y}+\boldsymbol{U})}\right] \cdot (h(y_i + 0.5) - h(y_i - 0.5)), \tag{20}$$

where the approximation in (19) is obtained by assuming the partial derivative $\frac{\partial}{\partial z_i} \ell(\boldsymbol{Z})$ is uncorrelated with $\frac{\partial}{\partial y_i} h(y_i + U_i)$. This would hold, for example, if $\ell$ were locally linear around $h(\boldsymbol{y})$ such that its derivative is the same for any possible perturbed value $h(\boldsymbol{y} + \boldsymbol{u})$.

Equation 20 corresponds to the following modification of backpropagation: the forward pass is computed in a standard manner (that is, evaluating $\ell(h(\boldsymbol{y} + \boldsymbol{u}))$ for a sampled instance $\boldsymbol{u}$), but in the backward pass we replace the derivative $\frac{\partial}{\partial y_i} h(y_i + u_i)$ with its expected value, $h(y_i + 0.5) - h(y_i - 0.5)$.

Consider a model where soft-rounding follows the encoder, $\boldsymbol{y} = s_\alpha(f(\boldsymbol{x}))$, and a factorial entropy model is used. The rate-distortion loss becomes

$$-\sum_i \mathbb{E}[\log_2 p_i(y_i + U_i)] + \lambda \mathbb{E}\left[d(\boldsymbol{x}, g(r_\alpha(\boldsymbol{y} + \boldsymbol{U})))\right]. \tag{21}$$

We can apply Equation 17 directly to the rate term to calculate the gradient of $\boldsymbol{y}$ (Figure 1B). For the distortion term we use Equation 20 where $r_\alpha$ takes the role of $h$. Interestingly, for the soft-rounding function and its inverse the expected derivative takes the form of a straight-through gradient estimate [7]. That is, the expected derivative is always 1.

Given a cumulative distribution $c_Y$ for $Y$, the density of $Z = s(Y) + U$ can be shown to be

$$p_{s(Y)+U}(y) = c_Y(s^{-1}(y) + 0.5) - c_Y(s^{-1}(y) - 0.5). \tag{22}$$

We use this result to parametrize the density of $Z$ (see Appendix E for details). Figure 1B illustrates such a model where $Y$ is assumed to have a logistic distribution.

# 5 Experiments

## 5.1 Models

We conduct experiments with two models: (a) a simple linear model and (b) a more complex model based on the hyperprior architecture proposed by Ballé et al. [6] and extended by Minnen et al. [23].

**The linear model** operates on 8x8 blocks similar to JPEG/JFIF [16]. It is implemented by setting the encoder $f$ to be a convolution with a kernel size of 8, a stride of 8, and 192 output channels. The decoder $g$ is set to the corresponding transposed convolution. Both are initialized independently with random orthogonal matrices [28]. For the density model we use the non-parametric model of Ballé et al. [6] but adjusted for soft-rounding (Appendix E).

**The hyperprior model** is a much stronger model and is based on the (non-autoregressive) "Mean & Scale Hyperprior" architecture described by Minnen et al. [23]. Here, the coefficients produced by a neural encoder $f$, $\boldsymbol{y} = f(\boldsymbol{x})$, are mapped by a second encoder $h$ to "hyper latents" $\boldsymbol{v} = h(\boldsymbol{y})$. Uniform noise is then applied to both sets of coefficients. A sample $\boldsymbol{w} = \boldsymbol{v} + \boldsymbol{u}_1$ is first transmitted and subsequently used to conditionally encode a sample $\boldsymbol{z} = \boldsymbol{y} + \boldsymbol{u}_2$. Finally, a neural decoder computes the reconstruction as $\hat{\boldsymbol{x}} = g(\boldsymbol{z})$. Following previous work, the conditional distribution is assumed to be Gaussian,

$$p(\boldsymbol{y} \mid \boldsymbol{w}) = \mathcal{N}\left(\boldsymbol{y}; m_\mu(\boldsymbol{w}), m_\sigma(\boldsymbol{w})\right), \tag{23}$$

where $m_\mu$ and $m_\sigma$ are two neural networks. When integrating soft quantization into the architecture, we center the quantizer around the mean prediction $m_\mu(\boldsymbol{w})$,

$$\boldsymbol{y} = s_\alpha(f(\boldsymbol{x}) - m_\mu(\boldsymbol{w})), \quad \boldsymbol{z} = \boldsymbol{y} + \boldsymbol{u}_2, \quad \hat{\boldsymbol{x}} = g(r_\alpha(\boldsymbol{z}) + m_\mu(\boldsymbol{w})). \tag{24}$$

and adjust the conditional density accordingly. This corresponds to transmitting the residual between $\boldsymbol{y}$ and the mean prediction $m_\mu(\boldsymbol{w})$ (soft rounded) across the uniform noise channel. As for the linear model, we use a non-parametric model for the density of $\boldsymbol{v}$.

We consider the following three approaches for each model:

**Uniform Noise + Quantization:** The model is trained with uniform noise but uses quantization for inference. This is the approach that is widely used in neural compression [e.g., 4, 5, 23, 36]. We refer to this setting as **UN + Q** or as the "test-time quantization baseline".

**Uniform Noise + Universal Quantization:** Here the models use the uniform noise channel during training as well as for inference, eliminating the train-test mismatch. We refer to this setting as **UN + UQ**. As these models have the same training objective as **UN + Q**, we can train a single model and evaluate it for both settings.

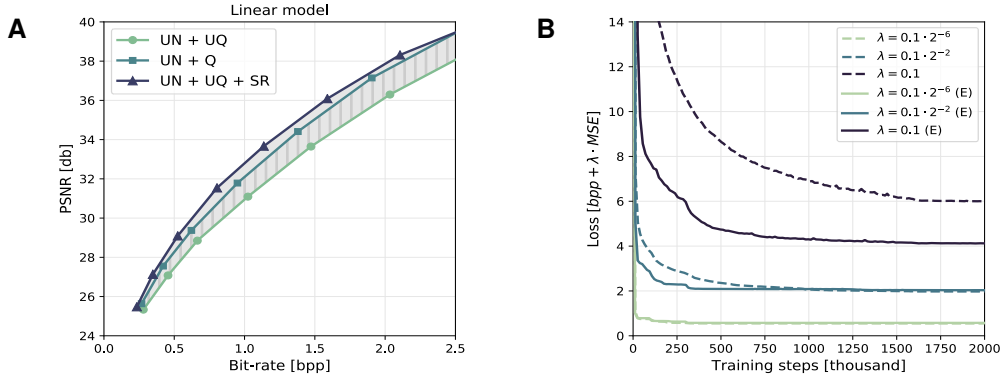

Figure 2: Comparison of linear models evaluated on the Kodak dataset [20]. **A**: The uniform noise channel (UN) by itself performs poorly in terms of PSNR. Using quantization at test time (Q) improves performance but leads to a train/test mismatch. Annealing soft-rounding (SR) towards quantization during training eliminates the mismatch and leads to improved performance. **B**: Effect of using expected gradients on the test loss of a linear model with $s_7$ and $r_{16}$. At high bit-rates (large $\lambda$) using gradient expectations (E) significantly improves our ability to train the model. At lower bit-rates, convergence is still faster but the final performance is similar.

**Uniform Noise + Universal Quantization + Soft Rounding:** Here we integrate a soft quantizer (Section 4.1) into the uniform noise channel (both during training and at test time), recovering the potential benefits of quantization while maintaining the match between training and test phases using universal quantization. We refer to this setting as **UN + UQ + SR**.

## 5.2 Traininig

The training examples are 256x256 pixel crops extracted from a set of 1M high resolution JPEG images collected from the internet. The images' initial height and width ranges from 3,000 to 5,000 pixels but images were randomly resized such that the smaller dimension is between 533 and 1,200 pixels before taking crops.

We optimized all models for mean squared error (MSE). The Adam optimizer [19] was applied for 2M steps with a batch size of 8 and a learning rate of $10^{-4}$ which is reduced to $10^{-5}$ after 1.6M steps. For the first 5,000 steps only the density models were trained and the learning rates of the encoder and decoder transforms were kept at zero. The training time was about 30 hours for the linear models and about 60 hours for the hyperprior models on an Nvidia V100 GPU.

For the hyperprior models we set $\lambda = 2^i$ for $i \in \{-6, \cdots, 1\}$ and decayed it by a factor of $\frac{1}{10}$ after 200k steps. For the linear models we use slightly smaller $\lambda = 0.4 \cdot 2^i$ and reduced it by a factor of $\frac{1}{2}$ after 100k steps and again after 200k steps.

For soft rounding we linearly annealed the parameter $\alpha$ from 1 to 16 over the full 2M steps. At the end of training, $\alpha$ is large enough that soft rounding gives near identical results to rounding.

## 5.3 Results

We evaluate all models on the Kodak [20] dataset by computing the rate-distortion (RD) curve in terms of bits-per-pixel (bpp) versus peak signal-to-noise ratio (PSNR).

In Figure 2A we show results for the linear model. When comparing the **UN + UQ** model which uses universal quantization to the test-time quantization baseline **UN + Q**, we see that despite the train-test mismatch using quantization improves the RD-performance at test-time (hatched area). However, looking at **UN + UQ + SR**, we obtain an improvement in terms of RD performance (shaded area) over the test-time quantization baseline.

In Figure 3A we can observe similar albeit weaker effects for the hyperprior model. There is again a performance gap between **UN + Q** and **UN + UQ**. Introducing soft rounding again improves the RD performance, outperforming the test-time quantization baseline at low bitrates. The smaller difference

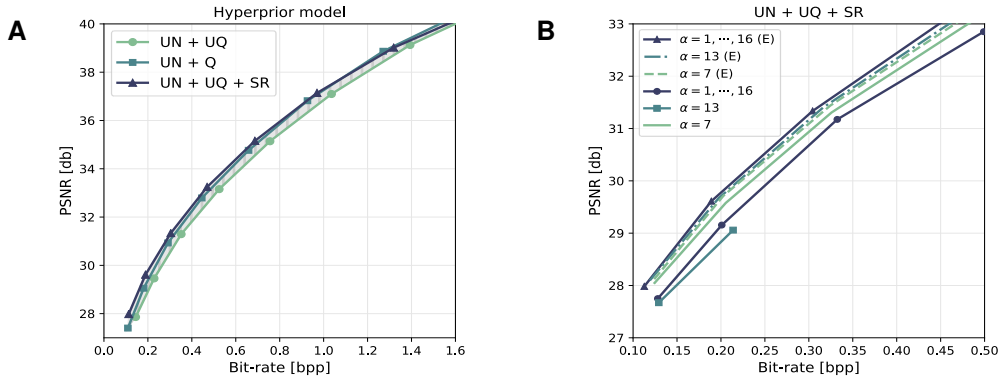

Figure 3: Evaluation of universal quantization and soft-rounding on a hyperprior model. **A**: As for the linear model (Figure 2) we find that test-time quantization (Q) beneficial compared to uniform noise, but that annealed soft-rounding (SR) improves performance at low bitrates. **B**: We find that expected gradients (E) improve the performance of models with soft-rounding, especially for large values of the parameter ($\alpha > 7$) – both when it is fixed ($\alpha = 13$) or annealed ($\alpha = 1, \cdots, 16$) during training.

can be explained by the deeper networks' ability to imitate functionality otherwise performed by soft-rounding. For example, $r_\alpha$ has a denoising effect which a powerful enough decoder can absorb.

In Figure 2B we illustrate the effect of using expected gradients on the linear model. We did not observe big differences when using the same $\alpha$ with $s_\alpha$ and $r_\alpha$ (not shown). However, using $s_7$ and $r_{16}$ we saw significant speedups in convergence and gaps in performance at high bitrates.

For the hyperprior model we find expected gradients beneficial both in terms of performance and stability of training. In Figure 3B, we consider the **UN + UQ + SR** setting using either the linear schedule $\alpha = 1, \cdots, 16$ or alternatively a fixed $\alpha \in \{7, 13\}$, with and without expected gradients. We found that for $\alpha > 7$, the models would not train stably (especially at the higher bitrates) without expected gradients (both when annealing $\alpha$ and for fixed $\alpha = 13$) and obtained poorer performance.

In summary, for both the linear model and the hyperprior we observe that despite a train-test mismatch, the effect of the quantization is positive (**UN + Q** vs **UN + UQ**), but that further improvements can be gained by introducing soft rounding (**UN + UQ + SR**) into the uniform noise channel. Furthermore we find that expected gradients are helpful to speed up convergence and stabilize training.

## 6 Conclusion

The possibility to efficiently communicate samples has only recently been studied in information theory [10, 11] and even more recently been recognized in machine learning [15, 13]. We connected this literature to an old idea from rate-distortion theory, uniformly dithered or universal quantization [27, 29, 37, 35], which allows us to efficiently communicate a sample from a uniform distribution. Unlike more general approaches, universal quantization is computationally efficient. This is only possible because it considers a constrained class of distributions, as shown in Lemma 1.

Intriguingly, universal quantization makes it possible to implement an approach at test time which was already popular for training neural networks [4]. This allowed us to study and eliminate existing gaps between training and test losses. Furthermore, we showed that interpolating between the two approaches in a principled manner is possible using soft-rounding functions.

For ease of training and evaluation our empirical findings were based on MSE. We found that already here a simple change can lead to improved performance, especially for models of low complexity. However, *generative compression* [26, 2] may benefit more strongly from compression without quantization. Theis et al. [31] showed that uniform noise and quantization can be perceptually very different, suggesting that adversarial and other perceptual training losses may be more sensitive to a mismatch between training and test phases. Roberts [27] found that replacing quantization with dithered quantization can improve picture quality when applied directly to grayscale pixels. Similarly, we find that reconstructions of the linear model have visible blocking artefacts when using quantization, as would be expected given the model's similarity to JPEG/JFIF [16]. In contrast,

universal quantization masks the blocking artefacts almost completely at the expense of introducing grain (Appendix G).

Finally, here we only studied one-dimensional uniform dither. Two generalizations are discussed in Appendix C and may provide additional advantages. We hope that our paper will inspire work into richer classes of distributions which are easy to communicate in a computationally efficient manner.

## Broader Impact

Poor internet connectivity and high traffic costs are still a reality in many developing countries [3]. But also in developed countries internet connections are often poor due to congestion in crowded areas or insufficient mobile network coverage. By improving compression rates, neural compression has the potential to make information more broadly available. About 79% of global IP traffic is currently made up of videos [17]. This means that work on image and video compression in particular has the potential to impact a lot of people.

Assigning fewer bits to one image is only possible by simultaneously assigning more bits to other images. Care needs to be taken to make sure that training sets are representative. Generative compression in particular bears the risk of misrepresenting content but is outside the scope of this paper.

## Acknowledgments

We would like to thank Johannes Ballé for helpful discussions and valuable comments on this manuscript. This work was performed and funded by Google.

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
