[Supplementary Material]

## Appendix A: Notation

| Example | Description |
|---|---|
| $z$ | Scalar |
| $\boldsymbol{z}$ | Vector |
| $Z$ | Continuous random variable |
| $K$ | Discrete random variable |
| $\boldsymbol{Z}$ | Continuous random vector |
| $p(\boldsymbol{z})$ | Continuous PDF |
| $P(k)$ | Discrete PMF |
| $H[K]$ | Discrete entropy |
| $h[Z]$ | Differential entropy |
| $\mathbb{E}[Z]$ | Expectation |
| $D_{\mathrm{KL}}[q_{\boldsymbol{Z}\mid\boldsymbol{x}} \mid\mid p_{\boldsymbol{Z}}]$ | Kullback-Leibler divergence |
| $D_{\mathrm{TV}}[q, p]$ | Total variation distance |

## Appendix B: Computational complexity of reverse channel coding

Existing algorithms for lossy compression without quantization communicate a sample by simulating a large number of random variables $Z_n \sim p$ and then identifying an index $N^*$ such that $Z_{N^*}$ is distributed according to $q$, at least approximately in a total variation sense [e.g., 3, 4]. Here we show that no polynomial time algorithm exists which achieves this, assuming $RP \neq NP$, where $RP$ is the class of randomized polynomial time algorithms.

Our result depends on the results of Long and Servedio [6, Theorem 13], who showed that simulating *restricted Boltzmann machines* (RBMs) [8] approximately is computationally hard. For completeness, we repeat it in slightly weakened but simpler form here:

**Theorem 1.** *If $RP \neq NP$, then there is no polynomial-time algorithm with the following property: Given as input $\boldsymbol{\theta} = (\boldsymbol{A}, \boldsymbol{a}, \boldsymbol{b})$ such that $\boldsymbol{A}$ is an $M \times M$ matrix, the algorithm outputs an efficiently evaluatable representation of a distribution whose total variation distance from an RBM with parameters $\boldsymbol{\theta}$ is at most $\frac{1}{12}$.*

Here, an *efficiently evaluatable representation* of a distribution $q$ is defined as a Boolean function

$$f : \{0,1\}^N \to \{0,1\}^M \tag{1}$$

with the following two properties. First, $f(\boldsymbol{B}) \sim q$ if $\boldsymbol{B}$ is a random vector of uniformly random bits. Second, $N$ and the function's computational complexity are bounded by a polynomial in $M$.

One might hope that having access to samples from a similar distribution would help to efficiently simulate an RBM. The following lemma shows that additional samples quickly become unhelpful as the Kullback-Leibler divergence between the two distributions increases.

**Lemma 1.** *Consider an algorithm which receives a description of an arbitrary probability distribution $q$ as input and is also given access to an unlimited number of i.i.d. random variables $\boldsymbol{Z}_n \sim p$. It outputs $\boldsymbol{Z} \sim \tilde{q}$ such that its distribution is approximately $q$ in the sense that $D_{TV}[\tilde{q}, q] \leq 1/12$. If $RP \neq NP$, then there is no such algorithm whose time complexity is polynomial in $D_{KL}[q \mid\mid p]$.*

*Proof.* Let $\boldsymbol{z} \in \{0,1\}^M$ be a binary vector and let

$$q(\boldsymbol{z}) = \frac{1}{Z} \sum_{\boldsymbol{h} \in \{0,1\}^M} \exp(\boldsymbol{a}^\top \boldsymbol{z} + \boldsymbol{h}^\top \boldsymbol{A} \boldsymbol{z} + \boldsymbol{b}^\top \boldsymbol{h}) \tag{2}$$

be the probability distribution of an RBM with normalization constant $Z$ and parameters $\boldsymbol{a}, \boldsymbol{b} \in \mathbb{R}^M$ and $\boldsymbol{A} \in \mathbb{R}^{M \times M}$. Further, let $p(\boldsymbol{z}) = 2^{-M}$ be the uniform distribution. Then

$$D_{\mathrm{KL}}[q \mid\mid p] = \sum_{\boldsymbol{z} \in \{0,1\}^M} q(\boldsymbol{z}) \log_2 \frac{q(\boldsymbol{z})}{2^{-M}} = M - H[q] \leq M. \tag{3}$$

If there is an algorithm which generates an approximate sample from an RBM's distribution $q$ in a number of steps which is polynomial in $D_{\mathrm{KL}}[q \mid\mid p]$, then its computational complexity is also

bounded by a polynomial in $M$. In that time the algorithm can take into account at most $N$ random variables $\boldsymbol{Z}_n$ where $N$ is polynomial in $M$, that is, $N = \psi(M)$ for some polynomial $\psi$. Since the input random variables are independent and identical, we can assume without loss of generality that the algorithm simply uses the first $N$ random variables. The $N$ random variables correspond to an input of $M\psi(M)$ uniformly random bits. Note that $M\psi(M)$ is still polynomial in $M$.

However, Theorem 1 states that there is no such polynomial time algorithm if $RP \neq NP$. $\qquad\square$

## Appendix C: Generalizations of universal quantization

Figure 1: A visualization of an example of the generalized uniform noise channel which can be implemented efficiently. Blue dots represent a lattice and black lines indicate corresponding Voronoi cells. The black dot corresponds to the coefficients $\boldsymbol{y}$ and the orange dots are realizations of the random variable $\boldsymbol{y} + \boldsymbol{U}$.

While the approach discussed in the main text is statistically and computationally efficient, it only allows us to communicate samples from a simple uniform distribution. We briefly discuss two possible avenues for generalizing this approach.

One such generalization to lattice quantizers was already discussed by Ziv [10]. Let $\Lambda$ be a lattice and $Q_\Lambda(\boldsymbol{y})$ be the nearest neighbor of $\boldsymbol{y}$ in the lattice. Further let $\mathcal{V}$ be a Voronoi cell of the lattice and $\boldsymbol{U} \sim U(\mathcal{V})$ be a random vector which is uniformly distributed over the Voronoi cell. Then [9, Theorem 4.1.1]

$$Q_\Lambda(\boldsymbol{y} - \boldsymbol{U}) + \boldsymbol{U} \sim \boldsymbol{y} + \boldsymbol{U}. \tag{4}$$

An example is visualized in Figure 1. For certain lattices and in high dimensional spaces, $\boldsymbol{U}$ will be distributed approximately like a Gaussian [9]. This means universal quantization could be used to approximately simulate an additive white Gaussian noise channel.

Another possibility to obtain Gaussian noise would be the following. Let $S$ be a positive random variable independent of $Y$ and $U \sim U([-0.5, 0.5))$. We assume that $S$ like $U$ is known to both the encoder and the decoder. It follows that

$$(\lfloor y/S - U \rceil + U) \cdot S \sim y + SU' \tag{5}$$

for another uniform random variable $U'$. If $G \sim \Gamma(\text{\small 3/2}, \text{\small 1/2})$ and $S = 2\sigma\sqrt{G}$, then $SU'$ has a Gaussian distribution with variance $\sigma^2$ [7]. More generally, this approach allows us to implement any noise which can be represented as a uniform scale mixture. However, the average number of bits required for transmitting $K = \lfloor y/S - U \rceil$ can be shown to be (Appendix B)

$$H[K \mid U, S] = I[Y, (Z, S)] \geq I[Y, Z], \tag{6}$$

where $Z = Y + SU'$. This means we require more bits than we would like to if all we want to transmit is $Z$. However, if we consider $(Z, S)$ to be the message, then again we are using only as many bits as we transmit information.

## Appendix D: Differentiability of soft-rounding

For $\alpha > 0$, we defined a soft rounding function as

$$s_\alpha(y) = \lfloor y \rfloor + \frac{1}{2}\frac{\tanh(\alpha r)}{\tanh(\alpha/2)} + \frac{1}{2}, \quad \text{where} \quad r = y - \lfloor y \rfloor - \frac{1}{2}. \tag{7}$$

The soft-rounding function is differentiable everywhere. First, we show that the derivative exists at $0$. The right derivative of $s_\alpha$ at $0$ exists and is given by

$$\lim_{\varepsilon \downarrow 0} \frac{s_\alpha(\varepsilon) - s_\alpha(0)}{\varepsilon} = \lim_{\varepsilon \downarrow 0} \frac{1}{\varepsilon}\left( \lfloor \varepsilon \rfloor - \frac{1}{2}\frac{\tanh(\alpha(\varepsilon - \lfloor \varepsilon \rfloor - \frac{1}{2}))}{\tanh(\alpha/2)} - \frac{1}{2}\frac{\tanh(-\alpha/2)}{\tanh(\alpha/2)} \right) \tag{8}$$

$$= \lim_{\varepsilon \downarrow 0} \frac{1}{\varepsilon}\left( \frac{1}{2}\frac{\tanh(\alpha(\varepsilon - \frac{1}{2}))}{\tanh(\alpha/2)} - \frac{1}{2}\frac{\tanh(-\alpha/2)}{\tanh(\alpha/2)} \right) \tag{9}$$

$$= \lim_{\varepsilon \downarrow 0} \frac{\tanh(\alpha(\varepsilon - \frac{1}{2})) - \tanh(-\alpha/2)}{2\varepsilon \tanh(\alpha/2)} \tag{10}$$

$$= \frac{1}{2\tanh(\alpha/2)} \lim_{\varepsilon \downarrow 0} \frac{\tanh(\alpha\varepsilon - \alpha/2) - \tanh(-\alpha/2)}{\varepsilon} \tag{11}$$

$$= \frac{1}{2\tanh(\alpha/2)} \left. \frac{\partial}{\partial x} \tanh(\alpha x - \alpha/2) \right|_{x=0} \tag{12}$$

$$= \frac{\alpha}{2}\frac{\tanh'(-\alpha/2)}{\tanh(\alpha/2)}. \tag{13}$$

Similarly, the left derivative at $0$ exists and is given by

$$\lim_{\varepsilon \uparrow 0} \frac{s_\alpha(\varepsilon) - s_\alpha(0)}{\varepsilon} = \lim_{\varepsilon \uparrow 0} \frac{1}{\varepsilon}\left( \lfloor \varepsilon \rfloor + \frac{1}{2}\frac{\tanh(\alpha(\varepsilon - \lfloor \varepsilon \rfloor - \frac{1}{2}))}{\tanh(\alpha/2)} - \frac{1}{2}\frac{\tanh(-\alpha/2)}{\tanh(\alpha/2)} \right) \tag{14}$$

$$= \lim_{\varepsilon \uparrow 0} \frac{1}{\varepsilon}\left( -1 + \frac{1}{2}\frac{\tanh(\alpha(\varepsilon + \frac{1}{2}))}{\tanh(\alpha/2)} - \frac{1}{2}\frac{\tanh(-\alpha/2)}{\tanh(\alpha/2)} \right) \tag{15}$$

$$= \lim_{\varepsilon \uparrow 0} \frac{-2\tanh(\alpha/2) + \tanh(\alpha(\varepsilon + \frac{1}{2})) + \tanh(\alpha/2)}{2\varepsilon \tanh(\alpha/2)} \tag{16}$$

$$= \frac{1}{2\tan(\alpha/2)} \lim_{\varepsilon \uparrow 0} \frac{\tanh(\alpha\varepsilon - \alpha/2) - \tanh(\alpha/2)}{\varepsilon} \tag{17}$$

$$= \frac{\alpha}{2}\frac{\tanh'(-\alpha/2)}{\tanh(\alpha/2)}. \tag{18}$$

Since the left and right derivatives are equal, $s_\alpha$ is differentiable at $0$. Since $s_\alpha(y + 1) = s_\alpha(y) + 1$, the derivative also exists for other integers and it is easy to see that $s_\alpha$ is differentiable for $y \notin \mathbb{Z}$. Hence, $s_\alpha$ is differentiable everywhere.

## Appendix E: Adapting density models

For the rate term we need to model the density of $f(\boldsymbol{x}) + \boldsymbol{U}$. When $\boldsymbol{y} = f(\boldsymbol{x})$ is assumed to have independent components, we only need to model individual components $Y + U$. Following Ballé et al. [2], we parameterize the model through the cumulative distribution $c_Y$ of $Y$, as we have

$$p_{Y+U}(y) = c_Y(y + 0.5) - c_Y(y - 0.5). \tag{19}$$

We can generalize this to model the density of $s(Y) + U$, where $s : \mathbb{R} \to \mathbb{R}$ is an invertible function. Since $c_{s(Y)} = c_Y(s^{-1}(y))$, we have

$$p_{s(Y)+U}(y) = c_Y(s^{-1}(y) + 0.5) - c_Y(s^{-1}(y) - 0.5). \tag{20}$$

This means we can easily adjust a model for the density of $f(\boldsymbol{X})$ to model the density of $s_\alpha(f(\boldsymbol{X})) + \boldsymbol{U}$. In addition to being a suitable density, creating an explicit dependency on $\alpha$ has the added advantage of automatically adapting the density if we choose to change $\alpha$ during training.

# Appendix F: Additional experimental results

Figure 2: Additional results for the linear model evaluated on the Kodak dataset [1]. **A**: The linear model as described in the main text but instead of a random orthogonal initialization, the linear transforms are initialized to the ones used by JPEG/JFIF [5]. That is, a YCC color transformation followed by a DCT for the encoder and corresponding inverses for the decoder. **B**: The linear model orthogonally initialized as in the main text but evaluated in terms of MSE instead of PSNR. **C**: The same linear model (orthogonally initialized, trained with respect to MSE) evaluated in terms of MS-SSIM.

Figure 3: Additional results for the hyperprior model evaluated on the Kodak dataset [1]. **A**: The hyperprior model from the main text but evaluated in terms of MSE instead of PSNR. **B**: The same model evaluated in terms of MS-SSIM. **C**: The effect of expected gradients shown for the full bitrate range.

## Appendix G: Qualitative results

Below we include reconstructions of images from the Kodak dataset [1] for the three approaches **UN + UQ**, **UN + Q**, and **UN + UQ + SR** trained with the same trade-off parameter $\lambda$. We chose a low bit-rate to make the differences more easily visible.

For the linear model (Figures 4-6), reconstructions using **UN + Q** and **UN + UQ + SR** have visible blocking artefacts as would be expected given their similarity to JPEG/JFIF [5]. **UN + UQ** masks the blocking artefacts almost completely at the expense of introducing grain.

For the hyperprior model (Figures 7-9), we noticed a tendency of **UN + Q** to produce grid artefacts which we did not observe using **UN + UQ + SR**.

**UN + UQ**, bpp: 0.762 (113%), PSNR: 32.79

**UN + Q**, bpp: 0.672 (100%), PSNR: 34.33

**UN + UQ + SR**, bpp: 0.562 (83%), PSNR: 33.60

Figure 4: Linear model, kodim03

**UN + UQ**, bpp: 0.836 (111%), PSNR: 32.22

**UN + Q**, bpp: 0.750 (100%), PSNR: 33.09

**UN + UQ + SR**, bpp: 0.623 (83%), PSNR: 32.54

Figure 5: Linear model, kodim16

**UN + UQ**, bpp: 0.778 (115%), PSNR: 33.05

**UN + Q**, bpp: 0.674 (100%), PSNR: 34.71

**UN + UQ + SR**, bpp: 0.572 (84%), PSNR: 33.98

Figure 6: Linear model, kodim23

**UN + UQ**, bpp: 0.091 (143%), PSNR: 29.36

**UN + Q**, bpp: 0.063 (100%), PSNR: 28.78

**UN + UQ + SR**, bpp: 0.059 (93%), PSNR: 29.55

Figure 7: Hyperprior model, kodim02

**UN + UQ**, bpp: 0.099 (137%), PSNR: 29.31

**UN + Q**, bpp: 0.073 (100%), PSNR: 28.82

**UN + UQ + SR**, bpp: 0.072 (99%), PSNR: 29.56

Figure 8: Hyperprior model, kodim15

**UN + UQ**, bpp: 0.156 (130%), PSNR: 27.11

**UN + Q**, bpp: 0.120 (100%), PSNR: 26.75

**UN + UQ + SR**, bpp: 0.123 (102%), PSNR: 27.08

Figure 9: Hyperprior model, kodim21