[Reviews · NeurIPS 2020]

Review 1

Summary and Contributions: This paper studies lossy compression technique for encoders. Unlike popular previous strategy to train the encoder with a differentiable approximation of the hard quantization by adding uniform noise, this paper first proposes to use universal quantization in both training and testing phase, which avoids the train-test mismatch issue. To deal with cases where one still wants to use quantization for training, the authors propose to use a smooth approximation to quantizing function based on universal quantization scheme, and provide a method to reduce the variance of its gradient computation. Finally, the authors provide experiments on Kodak dataset showing that with the proposed strategy, we can get lower PSNR and test loss.

Strengths: 1. The topic is useful in practice with many applications. It's will be interesting for many communities. 2. The ideas of adopting universal quantization in neural compression and replacing quantizing functions with a smooth surrogate are novel.

Weaknesses: The motivation is not clear enough for me. In other words, it seems that the result does not resolve the problem addressed. On the upside, it is absolutely a good attempt to apply universal quantization to this compression problem. However, the main concern is the effectiveness of this approach in practice. From Figure 2 and Figure 3, we see that using solely the proposed universal quantization (UN+UQ) is much worse than prior method directly using hard quantization (UN+Q). And, the performance of using additional soft rounding (UN+UQ+SR) is just slightly better than UN+Q, especially on the complicated model where the performances are almost the same. However, SR method introduces two more parameters (\alpha for s and r) which is harder to tune, while UN+Q is simpler. One of the major motivations of using UQ is to avoid the train-test mismatch, but from the experiments mentioned above, it performs even worse. This actually undermines the motivation. Additionally, the experiemnts are conducted only on one dataset, which makes the results less convincing. Hence, I would say that though applying UQ to encoder compression is new, the improvement of the paper is more or less marginal compared to previous work.

Correctness: Yes.

Clarity: Yes.

Relation to Prior Work: Yes.

Reproducibility: Yes

Additional Feedback: 1. It seems that function h is not defined consistently, e.g. eqn (7) and line 182. 2. I believe the soft rounding strategy should not be restricted to the UN and UQ scheme. Can we directly apply it to hard quantization in both training and testing, e.g. Q+SR? This is simpler and seems natural, which can also fix ''train-test mismatch''. 3. How is \alpha in s_\alpha and r_\alpha chosen, for example for the experimental results provided? 4. What is s in line 204? What does figure 1B try to say? I see that the green curve is far from the true gradient. 5. For the gradient variance reduction part, is upadting using the expectation driven by your empirical observation? Has similar idea shown in any previous work? ========================== After reading the rebuttal, I'm still not convinced by the improvement brought by the proposed method. Also, there surely are more datasets that can be used for testing this problem. Thus, I would keep the score as 5.


Review 2

Summary and Contributions: Neural network based compressors usually apply additive uniform noise during training as a proxy for the quantization that is performed during test-time. This creates a mismatch between the training and testing phases. This work proposes to instead apply universal quantization at test time thus eliminating the mismatch between training and test phases while maintaining a differentiable loss function. It is based on the fact that adding uniform noise to an input x is equivalent to subtracting a uniform random variable from x, rounding the result and then adding the same uniform random variable back. As a result, by sharing a random seed across the encoder and decoder we can easily implement universal quantization for neural network based compressors. The authors show that this is instance of a more general problem of efficiently communicating samples, which is computationally hard without distributional assumptions, but simple and practical for the uniform noise case. While this framework bypasses the need for quantization, the authors argue that there are still scenarios where one may desire hard quantization. As direct rounding does not allow for gradient based optimisation, the authors propose to instead use a soft rounding function that has a hyperparameter alpha; small values for alpha make the soft-rounding behave like an identity and large values of alpha make the soft-rounding behave like a hard one. Instead of directly applying their soft-rounding function on an input y, which is invertible and can lead to memorisation in the decoder, the authors further add uniform noise to the soft rounded value and then perform an MSE optimal reconstruction of the original y. Finally, the authors note that the variance of the gradients of the soft rounding function can be high for large values of alpha, and they propose a way to “marginalize” the randomness for a part of the gradient expression, which empirically leads to more stable optimisation. The authors then evaluate both universal quantization and universal quantization with soft-rounding to the Kodak dataset with a simple Linear model as well as a more flexible Mean & Scale Hyperpior model from prior literature. The results show that in general additive uniform noise with test-time hard quantization works better that universal quantization, however by incorporating soft-rounding they are able to improve, albeit slightly, upon the additive noise + test-time quantization setting.

Strengths: This work has contributions in two main themes which are relevant (for a part of) the NeurIPS community; compression with and without quantization. For the first point, the authors present soft rounding which allows for gradients to flow through, along with an approximate marginalisation of the noise in the gradients that can reduce their variance. For the second point, the authors employ the concept of universal quantization, which is simple, practical and easy to implement in existing frameworks. Both of these contributions can be valuable for a broad range of research in both data as well as model compression. The application of universal quantization as well as the approximate marginalisation for the gradients are novel contributions.

Weaknesses: While this paper is solid from a theoretical standpoint, I find that the empirical evaluation / experimental results are lacking. The premise of the paper seemed to imply that closing the gap between the training and test time behaviour of the algorithm, would be beneficial but unfortunately this does not seem to be the case. Additive uniform noise together with test-time hard-quantization seems to be better than universal quantization, implying that the mismatch of the train and test phases is not detrimental performance wise. Furthermore, while the addition of quantization to the universal quantization procedure seems to close the gap and improve upon the vanilla setting, the improvements seem to be small for the larger models which are typically employed, which makes me wonder whether for sufficiently large architectures universal quantization + soft rounding is necessary. Furthermore, evaluation on other datasets, such as BSD100 or Urban100, would make the experimental section of this work stronger. Finally, I would appreciate if the authors could elaborate a bit on the bias of the approximation done at eq. 19; does this assumption hold in practice? You could for example check the errors of your expression to the one where you use eq. 18 and average over multiple samples from the the original gradient (ideally at multiple stages of training).

Correctness: This work is technically correct.

Clarity: The writing of the paper could use some more work, as I often found it hard to follow. Up until page 3 the text reads nicely and is clear, but after that I noticed a couple of things: - The paragraph from line 141 to 146 is unclear and I would appreciate if the authors could explain it a bit better. What do you mean with the “single coefficient which is always zero”? Does such a case happen in practice? - At the intro of section 4 you mention that you will show that quantization is a limiting case of universal quantization if you allow for flexible encoders and decoders, but I don’t see any discussion on this part in the rest of the section. You rather introduce and work on round(round(y) + u) which is different from the round(y - u) + u of universal quantization. - What do you mean with the statement, “the prior is smooth enough to be approximately uniform in each interval“ at line 168-169? You mean that it is flat inside that interval? - At line 182 you mention that h is a differentiable function but later at line 187-188 you mention that you can evaluate the derivative of the expectation even when h is not differentiable (contradicting the previous assumption). Furthermore, I believe that more details about how the uniform noise + universal quantization + soft rounding setting works are appropriate: - Is the soft rounding at training time reverted to a hard-rounding at test time? - Where exactly is the universal quantization applied? At the intermediate result of soft_round(y) + u? - What happens when you employ soft rounding without universal quantization, i.e. soft_round(y) + u for training and just round(y) for testing?

Relation to Prior Work: The relation to prior work is generally good. Perhaps some interesting points for the authors is that the proposed rounding function seems to be similar / same as the one at [1] and that soft rounding of a noisy input has been also explored at [2]. [1] Differentiable Soft Quantization: Bridging Full-Precision and Low-Bit Neural Networks, Gong et al. [2] Relaxed Quantization for Discretized Neural Networks, Louizos et al.

Reproducibility: No

Additional Feedback: Typos and nitpicks: - Line 59-60, the correspondence to VAE is true for any distortion metric that corresponds to a log probability and not just the mean-squared error. - Line 116, “statistically efficiently” -> “statistically efficient“ - Line 137, “density that is” -> “density that” - Line 240, “Training” After reading the authors rebuttal I appreciate the authors summarisation of the contributions and clarifications; after reading them, I decided to keep my score the same. The work is definitely interesting and worthwhile to be considered for acceptance. The things that I would like to see in order to fully recommend acceptance would be the following: 1. You mention that universal quantization has the potential for big future improvements on other metrics, beyond PSNR, and mention a couple of other choices. While this could end up being the case, it could also end up being similar to the conclusions from the train / test mismatch we saw in this work. As a result, I would appreciate experimental evidence to satisfy these claims. 2. Exploration of the bias in the gradients introduced by performing the approximation at eq. 19. 3. Rephrasing the abstract and pitch of paper, as currently it gives the impression that the train / test gap is a problem, when in practice it is not. 4. _x0010_Improving the clarity of the paper, e.g, at the intro of section 4 you mention that you will show that quantization is a limiting case of universal quantization if you allow for flexible encoders and decoders, but I don’t see any discussion on this part in the rest of the section.


Review 3

Summary and Contributions: Authors describe two methods for handling quantization (a discontinuous operation) in learned systems, by 1) implementing a form of Ziv's Universal Quantization for transmitting noisy continuous values, and 2) introducing smoothed (and thus differentiable) quantizers. They examine performance of the first, and the combination of the two, on a simple linear coder, and on one recently published (Minnen et al, 2018).

Strengths: Focused and well-written, the paper builds on recent advances in deep CNN compression, in particular Balle et al 2017 and Minnen etal 2018. Those papers introduced a uniform noise approximation to the quantizer, so as to allow for a continuous and differentiable rate+distortion loss function during training, but then reverted to quantization for testing. Here the authors make use of Ziv's result to provide a direct encoding of the uniform noise values, thus allowing the test phase to be fully consistent with the training. i think this is a nice contribution (and was happy to learn about Ziv's result, which I'd not seen). The use, in addition, of a "softened" differentiable quantizer (as proposed in Agustsson 2019) leads to better performance.

Weaknesses: * I was surprised to see, after the authors touted advantages of using a consistent training and test implementations, that the results of the (UN+UQ) system were significantly worse than those of the "inconsistent" solution introduced by Balle et al 2017 (UN+Q). Only when the softened quantizer is added (UN+UQ+SR) do we see a relatively small improvement. Why? Are there potential ways to improve this? * This makes one wonder about how much the UQ noise matters. In particular, it would be instructive to see a comparison to (UN+SR). Given the previous comment, one might suspect this would lead to even better performance - and thus that the UQ methodology, despite its mathematical interest, is not of practical value. * Although I think it's important and interesting, this is a pretty heavy and narrowly-focused topic for the NeurIPS community.

Correctness: Math generally solid.

Clarity: Generally clearly written, especially for such a specialized technical topic.

Relation to Prior Work: Solid bibliography.

Reproducibility: Yes

Additional Feedback: >>> Added after reading author's feedback, and other reviewer comments: I appreciate the conceptual and mathematical points of the paper, and think they are valuable for the compression-interested sub-community at NeurIPS. I also appreciate the value of quantifying the train/test mismatch that arises from the uniform noise approximation (even thought it comes out relatively small). On the other hand, I see no evidence that "universal quantization has the potential to lead to much bigger improvements in the future." And consistent with R2, I think the authors should re-frame the initial statement (in abstract/intro) of the contributions. Overall, I did not find the feedback or other reviews significantly altered my view of the paper, which is above threshold. In any case, I hope the authors will find our comments helpful in improving the work.


Review 4

Summary and Contributions: This paper proposes an approach to account for quantization noise based on Ziv's universal quantization principles.

Strengths: The addressed problem is relevant and timely and may have potentially a broad impact. The paper is very clear and very well written. The mathematical explaination is also clear, even though it owes a lot to Ziv.

Weaknesses: Experiments with models less complex than scalable hyperpriors and more complex than linear would have given a better understanding of the effectiveness of the technique. For example, the authors could experiment with a simple autoencoder with a residual encoder. It is not totally clear how to practically apply the propsoed scheme due to the lack of a section that explains how to apply it, which may limits the reprodicibility of teh results. This work is limited by definition to uniform noise. What is the extra training complexity introduced by the proposed method? can it be quantified ? The experiments are performed on one dataset only (altough large), which is limiting. Is the source code provided? There are some typos, like for example in the title of sec. 5.2 "Traininig"

Correctness: Yes, albeit stronger experimental evaluation is welcome.

Clarity: Very well written.

Relation to Prior Work: Prior work is discussed appropriately.

Reproducibility: No

Additional Feedback:

[Author Response · NeurIPS 2020]

Dear reviewers, thank you for your time to thoroughly read and review our paper. For your convenience, we briefly summarize our contributions again below before addressing some of your specific questions and criticisms.

You said "the addressed problem is relevant and timely" (R4), "useful in practice with many applications" (R1), with the paper being "focused and well-written" (R3), having "contributions which are relevant (for a part of) the NeurIPS community" (R2) and "interesting for many communities" (R1). You further acknowledged the following contributions:

- Demonstrating that a uniform noise (UN) channel can be implemented at test time (R1, R2, R3)
- Eliminating the train-test mismatch while maintaining a differentiable loss function (R1, R2, R3)
- Bridging compression with and without quantization by showing that quantization is a limiting case of a soft quantizer applied to the uniform noise channel (R2, R3)
- Reducing the variance of gradients by analytically integrating out the noise (R1, R2)

We want to highlight these additional contributions which you may have missed:

- *Proving that the general problem of communicating a sample from an arbitrary distribution is computationally hard* (Lemma 1, l.105). This has practical implications and makes the special case of uniform noise particularly interesting. Beyond compression, our lemma shows that sampling from a distribution is generally hard even with access to samples from a related distribution.
- Quantifying for the first time the gap between train and test losses in the approach of Ballé et al. (see below)

**The proposed approach only marginally improves PSNR for hyperprior models. (R1, R2, R3)** While the benefits of our approach in terms of PSNR are modest for the complex hyperprior model, we observe significant improvements for the linear model (Figure 2A). Note that in compression *lightweight models are very relevant in practice*.

Secondly, *our empirical results allow us for the first time to quantify the gap between training and test losses*. Until now, nobody knew to which extent a train-test mismatch hurts (or helps) performance. That is, our results are interesting from a practical point of view even without any immediate improvements in performance.

Finally, *universal quantization has the potential to lead to much bigger improvements in the future*. While MSE and PSNR are still the most widely used metrics, they are also flawed. Adversarial losses and advanced perceptual metrics will be more sensitive to the perceptual differences between quantization and noise. However, as these metrics are still an active area of research and can be difficult to train and evaluate, we believe it is right to focus on PSNR first.

**Why did you only evaluate on the Kodak dataset? (R1, R2, R4)** Unlike other tasks, compression results tend to generalize well across natural image datasets (e.g., Agustsson et al., 2017). This explains why evaluations on a single dataset, namely Kodak, are common practice (e.g., Theis et al., 2016; Ballé et al., 2018; Choi et al., 2019). We therefore decided to use the limited space to explore our conceptual contributions instead of providing results on other datasets.

**Can we fix the train-test mismatch by applying soft-rounding to hard quantization during both training and testing? (R1)** Combining hard quantization with soft-rounding would be equivalent to hard quantization, $s(\lfloor s(y) \rceil) = \lfloor y \rceil$, and thus would not be differentiable. Using soft-rounding without noise during training (Agustsson et al., 2017) would not create a bottleneck (since soft-rounding $s_\alpha$ is invertible for any finite $\alpha$). Agustsson et al. (2017) point out that "choosing the annealing schedule is crucial" (p.6) as annealing needs to be fast enough to avoid inversion, which then causes large gradients. We do not suffer from these problems due to the noise and our proposed variance reduction.

**What is Figure 1B trying to say? (R1)** Figure 1B shows an example where the derivatives of $h(y)$ can vary wildly for small changes in $y$. Using $h'(y + u)$ with sampled noise $u$ during training thus leads to gradients with high variance. On other hand, the expected derivative $\mathbb{E}_U[h'(y + U)]$ varies smoothly and thus leads to gradients of lower variance.

**Why does taking the expectation reduce variance? (R1)** Taking the expectation to reduce variance is a commonly used trick which can be motivated by the *law of total variance*, $\mathrm{Var}[\mathbb{E}[\Delta \mid U]] = \mathrm{Var}[\Delta] - \mathbb{E}[\mathrm{Var}[\Delta \mid U]] \leq \mathrm{Var}[\Delta]$.

**What do you mean with the "single coefficient which is always zero"? (R2)** The encoder outputs $\mathbf{y} = f(\mathbf{x})$. If $y_i = 0$ and the same noise is used everywhere (Choi et al., 2019), $\mathbf{z} = \mathbf{y} + u$, then the decoder could recover $\mathbf{y} = \mathbf{z} - z_i$.

**What is the added training complexity of the proposed method? (R4)** For the hyperprior model (UN + UQ + SR) we observe an increase in training times of 2-4% with variance reduction and less than 1% without variance reduction.

**Is the source code provided? (R4)** Source code will be made available.

**Related work on soft-rounding. (R2)** We'll add references to the mentioned work on soft-rounding.

**Do you mean the prior should be flat in the interval at l. 168-169? (R2)** That is correct, we will clarify the text.

**Where exactly is the universal quantization applied with soft-rounding? (R2)** We apply it to the result of soft-rounding, $s(y)$, i.e. $s(y) + u$ during training and universal quantization $\lfloor s(y) - u \rceil + u$ at test-time. See also l.254.

[Meta-Review · NeurIPS 2020]

*PROS: proposes to use universal quantization in both training and testing phase, which avoids the train-test mismatch issue, topic is useful in practice with many applications, clearly relevant contributions stated in the rebuttal *CONS: One of the major motivations is that the proposed method avoids the train-test mismatch, but from the experiments it performs worse. Lack of Exploration of the bias in the gradients introduced by performing the approximation at eq. 19 Meta-reviewer recommendations: The paper is borderlinle but I recommend acceptance. The authors presented a Very good rebuttal clearly stating the usefulness of the paper. I recommend the authors to try to explain why performance in the experiments is worse than approahes based on hard quantisation: Balle et al 2017 (UN+Q). My impression is that other techniques in the literature has not been as tuned as Balle's method, but this is only speculation. I also recommend the authors to note the reviewers' comments that the authors should soften the initial statement (in abstract/intro) of the contributions since there is no strong evidence that "universal quantization has the potential to lead to much bigger improvements in the future."